# A Review on Cement-Based Composites for Removal of Organic/Heavy Metal Contaminants from Water

**Vishvendra Pratap Singh [1], Rahul Vaish [1,\*] and El Sayed Yousef [2,3]**

1   School of Mechanical and Materials Engineering, Indian Institute of Technology Mandi, Mandi 175005, India
2   Research Center for Advanced Materials Science (RCAMS), King Khalid University, P.O. Box 9004, Abha 61413, Saudi Arabia
3   Physics Department, Faculty of Science, King Khalid University, P.O. Box 9004, Abha 61413, Saudi Arabia
\*   Correspondence: rahul@iitmandi.ac.in

**Abstract:** Building materials are traditionally known for their mechanical and structural properties. As environmental pollution has risen as a huge global issue, functional building materials with environmental remediation capabilities are the demand for the present time. In this context, cement and concrete with photocatalytic and adsorbent additives were explored for air and water remediation. The usage of functional building materials for self-cleaning and air cleaning is well documented and reviewed in earlier reports. This article gives an overview of the functional building material composites used for water remediation. Numerous different approaches, such as photocatalysis, adsorption, and antimicrobial disinfection, are discussed. Among all, photocatalysis for the degradation of organic compounds and antimicrobial effect has been the most studied method, with $TiO_2$ being the first choice for a photocatalyst. Furthermore, some reports illustrate the impact of photocatalytic filler on hydration and mechanical properties, which is important in case these are used in construction. Adsorption was most preferred for heavy metal removal from the water. This article rationalizes the current status and future scope of cement-based functional composites for water cleaning and discusses their use in water cleaning facilities or regular construction.

**Keywords:** photocatalysis; cement; adsorption; dye; ferroelectric

## 1. Introduction

Although water is a vital ingredient in our lives, its quality is continuously being degraded immensely as a result of industrialization and development. The uncontrolled discharge from various industries and households has decreased freshwater quality (rivers, underground water, ocean, etc.) [1]. Consequently, the availability of pure water is diminishing daily. Further, not only pollutants but various kinds of microorganisms (viruses, bacteria, protozoa, etc.) are also found in the contaminated water to which industrialized farming contributes heavily. These are responsible for some serious diseases such typhoid, cholera, hepatitis, etc. All these diseases can be life-threatening if not medicated properly.

Although many treatment processes such as chlorination, anaerobic and aerobic digestion, ozonation, adsorption, precipitation of heavy metals, and photocatalysis have been developed to counter this issue, there are still many challenges that are associated with each process [2–6].

Photocatalysis and adsorption are the two most studied techniques in this context for both air and water cleaning [7]. With respect to photocatalysis, the materials discussed most are $TiO_2$ and ZnO, mostly as nanosized particles [8,9]. Various research contributions can be found concerning the impact of these materials on the degradation of organic pollutants and also their antimicrobial effect, which is related to photocatalysis. The use of natural as well as artificial light for environmental remediation has become possible after the discovery of photocatalysis by Fujishima and Honda [10]. With further advances in the field, the potential application of photocatalysts along with construction and building

materials has been identified and explored after the 1990s. $TiO_2$ was the most exploited photocatalyst for this purpose [11].

Multiple endeavors aim at transferring photocatalytic water cleaning into application [12,13]. One of the questions that needs to be addressed is the support material on which these nanomaterials are immobilized [14,15]. Even though the nanoparticles show more degradation efficiency when they are suspended in water, fixing them should prevent the particles from remaining in the water because $TiO_2$ and $ZnO$ particles show phytotoxicity and likely cytotoxicity, they pose a risk to the environment and human health as well [16–18]. Polymers might not be a good choice because these degrade rapidly, especially when photoactive fillers are involved. Thus, leakage of the particles results. Ceramics, or in our case cement and naturally occurring clay materials, could be the right material as nanocatalyst support.

The primary function of cement, concrete, and other building materials is structural; however, its pervasiveness demands maintaining its aesthetic quality and integrity. In the present scenario, when environmental pollution is a major global concern, we have started thinking about environment-friendly engineered structures and buildings. During the earlier stages, the use of photocatalytic building materials was focused mostly on the removal of pollutants in the air and self-cleaning surfaces. The presence of $NO_x$, $SO_x$, and volatile organic compounds (VOCs) are highly undesirable in the air [19]. Cement and concrete-based photocatalytic composites and structures were found to significantly diminish all these pollutants in the air [20]. Similarly, the inclusion of a photocatalytic material in building materials such as cement can induce self-cleaning properties [21]. The literature also consists of only a few detailed reviews incorporating all the work done in this field [22–24]. However, these materials can also play a vital role in water detoxification. Similarly, adsorption is also an extensively studied water detoxification technique that is based on extracting the pollutant in the water onto the adsorbent's porous surface [25]. Cement, clay, and prepared composites of adsorbents can be used for water detoxification, e.g., in the form of filters [26,27].

This article reviews the role of functional construction materials on water cleaning (removal of pollutants and antimicrobial effects) based on relevant reports available in the literature. It will furthermore address if the different approaches could be feasible for the application in construction materials or if the strategies rather fit to localized water cleaning facilities. A schematic representing all explored pollutant removal capabilities of functional construction materials illustrated in this review is shown in Figure 1.

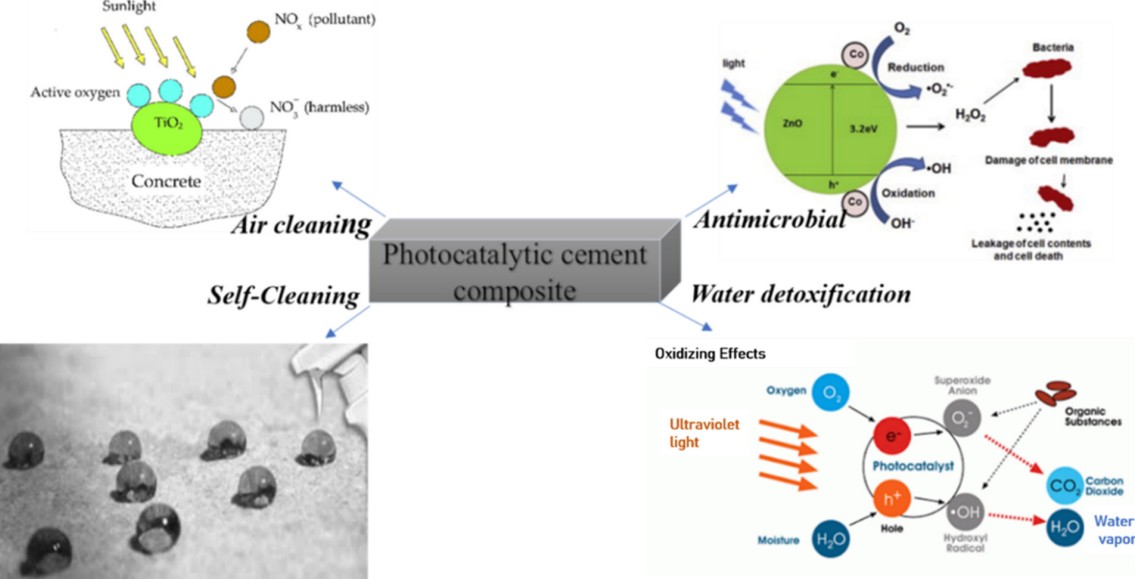

**Figure 1.** Various fields of application of functional construction materials.

## 2. Photocatalysis and Photocatalytic Materials

Semiconductor photocatalysis has now been studied for several decades and also successfully applied to air and water decontamination. In a typical photocatalytic reaction complex, organic molecules are broken down into $CO_2$ and $H_2O$ or simple hydrocarbons in the presence of a reaction catalyst (known as photocatalyst) [28]. Photocatalytic oxidation (PCO) has many advantages to offer over other available techniques for water treatment. The biggest one is that it mineralizes the pollutants rather than transferring them from one surface to another. Additionally, a photocatalyst is not consumed in the reaction and can be used for longer durations. This enhances the longevity of the process without the need to change the materials after each cycle. Although there are numerous known photocatalytic materials such as $TiO_2$, ZnO, $Bi_2WO_6$, etc. [29], among all, $TiO_2$ has been extensively studied and also adopted commercially due to its high efficiency and chemical stability [30]. The energy bandgap between the conduction and valence band is 3.20 eV. When incident photons provide sufficient energy, the electrons start migrating from the valence to conduction band, leading to the formation of holes in the valence band. These electron-hole pairs trigger the redox reaction in the system, and highly reactive free radicals ($-OH., O_2$) are generated. Ultimately, the target organic molecules are degraded by these free radicals into $CO_2$ and $H_2O$ or simple hydrocarbons. A schematic of the process is shown in Figure 2.

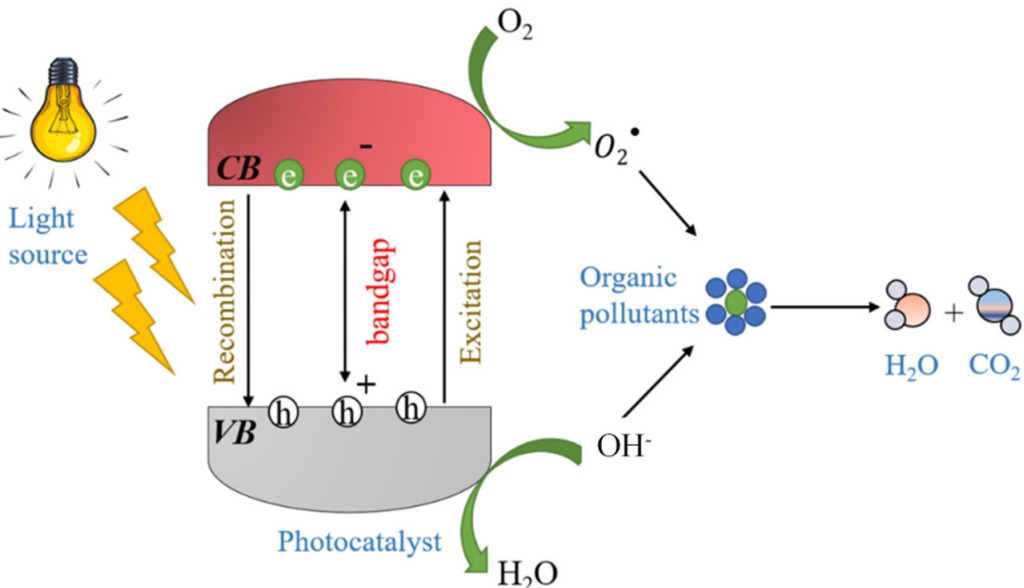

**Figure 2.** Schematic diagram of the photocatalytic degradation process.

After $TiO_2$, ZnO is the second-most used photocatalyst for the same purpose [31]. Although photocatalysis is an effective approach for removal of organic contaminants from waters, it is the most effective under UV light irradiation [8]. It limits its applicability in many practical applications. Specifically, photocatalytic building materials are often intended to operate under natural solar light that only contains 4% UV irradiation. Therefore, more visible-light-active photocatalyst should be explored that can be mixed with cement and other building materials to achieve appreciable degradation in solar light. The photocatalytic properties also give the materials antimicrobial, antiviral, and antifungal properties. Numerous reports claim that mixing photocatalytic materials ($TiO_2$ and ZnO) or carbonaceous material such as graphene can impart these activities to cement [32,33]. The mechanism of photocatalytic bacterial degradation is shown in Figure 3.

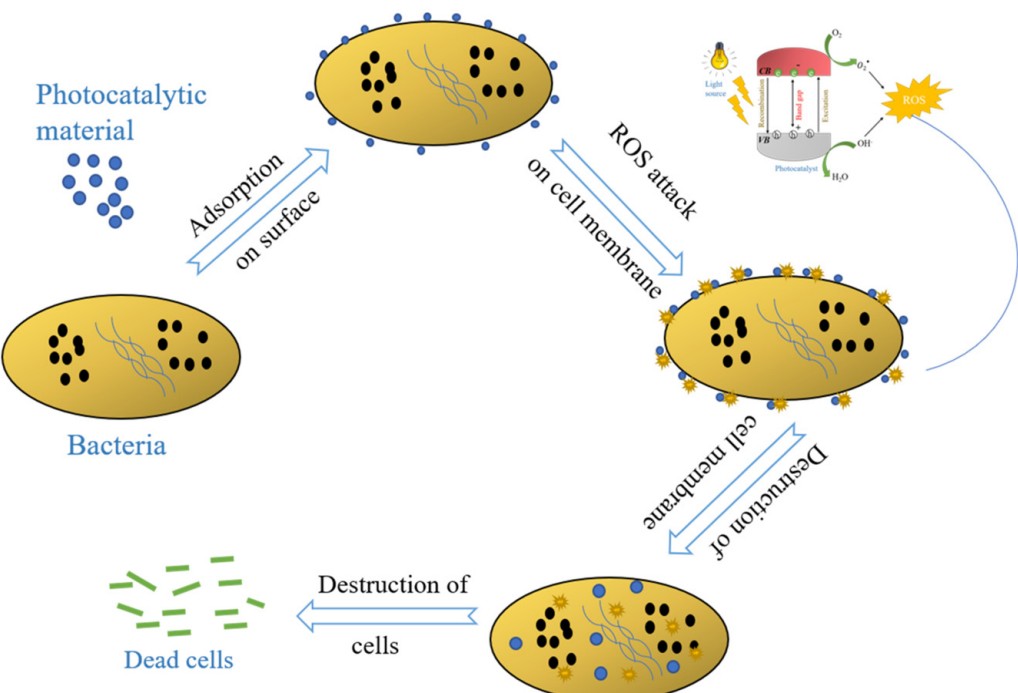

**Figure 3.** A schematic demonstrating various steps involved in photocatalytic bacterial disinfection.

As discussed earlier, during a typical photocatalysis process, highly reactive free radicals are generated in the system. These free radicals are responsible for bacterial disinfection by affecting multiple cell properties. Different mechanisms have been proposed by various researchers in the respective reports: (i) cell death due to oxidation of the intracellular coenzyme A (CoA) and (ii) decomposition of cell walls due to significant disorder in the cell permeability. Additionally, in one of the reports, Gogniat et al. suggested that adsorption of cells on $TiO_2$ is also correlated with the disinfection process. They proposed that loss of membrane integrity after being adsorbed on the catalyst was vital for the bactericidal effect of photocatalysis [34].

## 2.1. Photocatalysis-Based Water Detoxification Using Building Material Composites

A photocatalyst can be easily mixed with cement or concrete without greatly affecting its physical and chemical properties. Therefore, many reports can be found in the literature that claim photocatalytic air or water cleaning using composites of $TiO_2$ and structural materials [35]. Apart from economic reasons to use cement or clay as support for photocatalysts in water purification plants, applying these for self-cleaning structures such as bridges, etc., has become an interesting topic. In the literature, the use of photocatalytic materials along with building materials started in the 1990s. Initial endeavors were motivated toward maintaining the aesthetic quality of building materials; however, these composite materials were also later successfully tested and applied for environmental remediation. Most of the work described in the literature highlights the photocatalytic activity of the composite made using either $TiO_2$ or ZnO as filler. For example, aqueous solutions of various dyes such as methylene blue (MB), rhodamine B (RB), and some other phenolic compounds were used as a model pollutant [36].

Niessner et al. studied the performance of various photocatalysts. They implemented $TiO_2$ (Degussa P-25, Hombikat UV 100, anatase) and ZnO as filler in a matrix of white Portland cement [37]. Atrazine ((2-chloro-4-(ethylamino)-6-(isopropylamino)-s-triazine)), an herbicide used in crop cultivation, was taken as a model pollutant. Thus, this study also focuses on a generally relevant contaminant. The schematic of the experimental setup is shown in Figure 4.

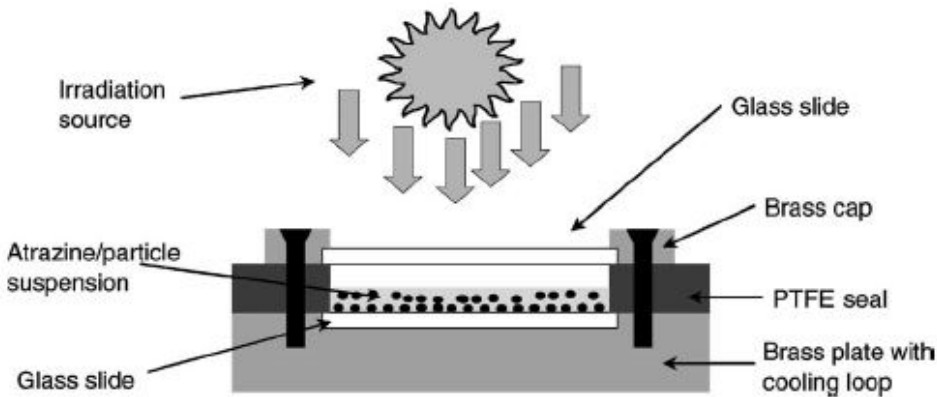

**Figure 4.** Schematic setup of an irradiation chamber (ref. [37]).

A solution of 1 μg/L concentration was prepared, and the degradation of atrazine using all cement composites was compared. Significant photocatalytic activity was observed with $TiO_2$-added samples; however, negligible activity was manifested by ZnO and anatase-$TiO_2$ samples under sunlight. Along with photocatalytic activities, some other interesting observations were also reported by the authors. The photocatalytic efficiency of composites was influenced by the particle size of semiconductor materials. Additionally, the activity was found to decrease with the aging of hardened cement pastes. The carbonation of the cement components is expected to be the reason for this. However, cement samples modified with Degussa P25 or Hombikat UV 100 showed much better degradation efficiency as compared to pure cement samples. Although the inclusion of photocatalytic filler can impart additional functionalities to cement, it is equally important to consider the mechanical and hydration characteristics of the cement composites. The authors took care of that and reported hydration kinetics and compressive strength of both modified and unmodified samples. The compressive strength of composites slightly improved for Degussa P25 or Hombikat UV and decreased for ZnO.

Improved cement hydration rate was also observed with the inclusion of $TiO_2$ (Degussa P25 and Hombikat UV 100). Therefore, this nicely illustrates that endeavors to add functional properties to structural materials with this approach can actually also be beneficial for the structural properties. The setup of this experiment considered a laboratory-scale approach with ideal conditions. Neppolian et al., however, conducted a parameter study of photocatalytic degradation of the organic dye reactive-yellow 17 with $TiO_2$ using cement a binder under sunlight [38]. They were able to illustrate the effect of a limited amount of influence factors that could affect the photocatalytic performance. Parameters such as the solution pH, temperature, and intensity of light were varied for the experiments. The mass of the cement–$TiO_2$ mixture (1:1 ratio) was varied from 100, 150, 200, 250 to 300 mg. It was observed that the degradation of dyes increased with increasing amount of catalyst up to 200 mg. It is due to the presence of more active sites that can produce more hydroxyl ions. Beyond that, degradation efficiency reduced increasing the catalyst amount. The reason suggested was an increment in turbidity of the suspension, which reduced the transmission of light in the system. These observations are important to consider while selecting a cement-based photocatalytic composite for environmental remediation. An optimum quantity of filler as well as matrix should be used for maximum performance. The effect of solution pH on the degradation was also studied in detail, and neutral pH was suggested best for the catalytic activity. Further, the influence of additives such as $H_2O_2$, $K_2S_2O_8$, and $Na_2CO_3$ was demonstrated. $Na_2CO_3$ was found unfavorable for the reaction while the other two ($H_2O_2$ and $K_2S_2O_8$) boosted the performance of the catalyst. The detrimental effect is in line with the findings by Niessner et al. discussed above, which illustrate the aging effect of carbonation [37].

It is essential that along with parameters such as the selection of catalytic materials, amount of catalyst used, etc., one can also adjust other reaction parameters, such as pH

of the solution, to achieve better efficiency with the same material. It also gives freedom to fix the composition at a level where other structural properties are best, and efficiency can be improved with such strategies. Another work demonstrating strategies to keep the structural properties intact even after imparting photocatalytic properties was introduced by T. Vulic and co-workers. They reported that the introduction of an inorganic–inorganic nanocomposite photocatalyst improves the compatibility of cement and photocatalyst, which is based on layered double hydroxides (LDHs) associated with $TiO_2$ [39]. LDHs can alter physical and chemical properties during synthesis as these materials represent one of the cement hydrations phases. Zinc (Zn) was chosen as LDH metal as it offers the additional advantage of being a photocatalytic and antimicrobial compound in the form of Ti–Zn–Al nanocomposite prepared from wet impregnation of $TiO_2$ onto Zn–Al layer double hydroxides. Ordinary Portland cement was used to fabricate two sets of samples: one control (not containing any additive) and other containing Ti–Zn–Al LDH as filler in the cement. After preparation, the physiochemical properties and photocatalytic activity of both samples were monitored.

Interestingly, the microhardness was almost doubled for the Ti–Zn–Al cement composite compared to that of the pure cement sample. The reason for such enhancement was stated to be the homogenous spreading of the fibrous phase to the composite surface. Furthermore, the photocatalytic activity of pure $TiO_2$, Zn–Al LDH, and Ti–Zn–Al LDH has been compared in the powdered form. The activity of pure $TiO_2$ noted highest among all. However, Ti–Zn–Al LDH powder also showed comparable activity during the degradation of MB. Additionally, the Ti–Zn–Al LDH powder was also introduced to the bulk cement in the form of a coating. The observations showed enhancement in the photocatalytic activity of cement with inclusion Ti–Zn–Al both in the bulk and coating forms. This report presented useful observations regarding how photocatalytic material can not only provide additional functionality to cement, but also improve its physicochemical properties using Ti–Zn–Al nanocomposites. However, the report does not compare the photocatalytic and physiochemical properties of $TiO_2$ cement composite with pure and prepared Ti–Zn–Al nanocomposite samples.

Elena et al. proposed a different approach for improving photocatalytic and hydration properties of cement–$TiO_2$ composites [40]. Photocatalytic cement paste was prepared using the sol–gel method. $TiO_2$ was admixed with cement to impart photocatalytic functionality using the sol–gel of synthesized $TiO_2$. The prepared composite was found to degrade MB in aqueous solutions successfully without degrading the mechanical properties. Based on scanning electron microscope (SEM) images of the composites, certain important points were noticed. $TiO_2$ particles were identified in the fibers of calcium silicate hydrate C-S-H gel matrix. Further, it is quite clear from the discussion that composites of $TiO_2$ with cement and concrete are quite efficient for organic pollutant degradation in water.

The activity of the catalyst can also be improved with more exposure of light on its surface. Lime et al. demonstrated $TiO_2$/autoclaved cellular concrete composite ($TiO_2$–ACC) has higher degradation efficiency than $TiO_2$ (P-25) in similar conditions [41]. The contaminants used for the degradation study were indigo carmine, diclofenac, and atrazine. Interestingly, almost 100% removal was obtained with the composites, while only 60% removal was obtained with P-25 and photolysis after 350 min.

The reason for such improvement in the activity of composites is its floating nature. Because of this, it floats on the surface, and higher surface area is exposed to receive more UV irradiation. Another interesting approach for enhancement of photodegradation efficiency is proposed by fabricating $TiO_2$/foam cement composites [42]. The porous $TiO_2$/foam composites were fabricated by mixing foaming-agent hydrogen peroxide and an early-age strengthening reagent of $Na_2SO_4$. The content of $TiO_2$ was varied from 0.1 to 0.3%, and five compositions were fabricated. The surface morphology of these composites is shown in Figure 5a–d.

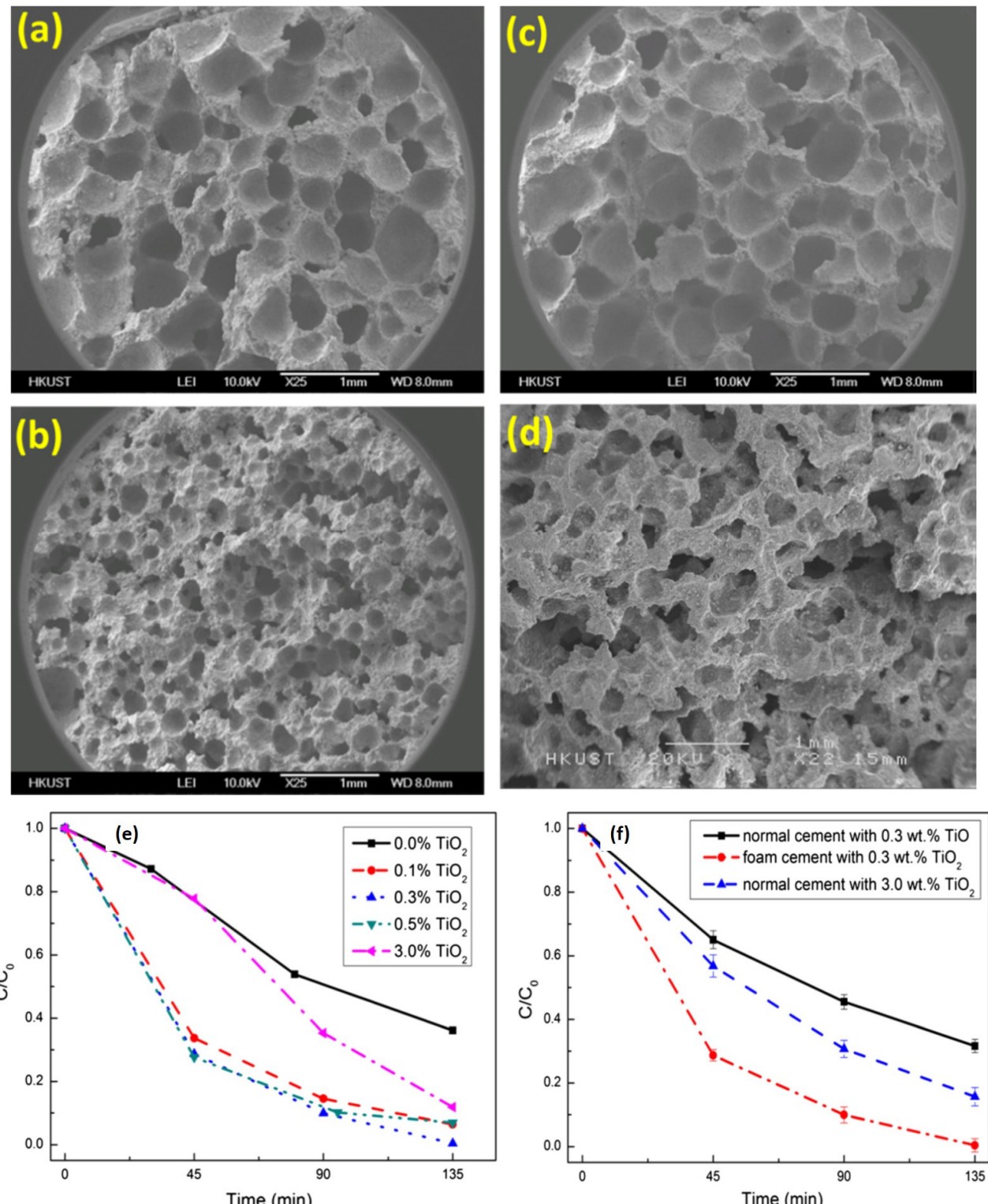

**Figure 5.** SEM micrographs of porous cement with TiO$_2$ of (**a**) 0.1 wt.%; (**b**) 0.3 wt.%; (**c**) 3.0 wt.%; (**d**) microstructure of the 0.3 wt.% TiO$_2$ modified foam cement; photodegradation of MB (**e**) foam cement with different amounts of TiO$_2$; (**f**) normal/foam cement with 0.3 wt.% TiO$_2$ and normal cement with 3.0 wt.% TiO$_2$. Ref. [42].

These results suggest that with the increment in the amount, the porosity of the foam cement slightly decreases. Additionally, the degradation was significantly higher in foam-based cement as compared to that of ordinary cement. A significant improvement of 80% was recorded with foam cement with a porosity of 56.41% and $TiO_2$ of 0.3 wt.% than that of normal cement with 0.3 wt.% $TiO_2$. The enhancement in photodegradation was attributed to the extended penetration of light in the foam. This ultimately caused quick activation of catalyst and, therefore, higher degradation activity, and a higher degradation rate was obtained. Jafari et al. used a sol–gel method to deposit thin films of $TiO_2$–$SiO_2$ nanohybrids [43]. The major highlight was the enhancement in photocatalytic activity after coating onto cement blocks. The cause of such an increase was the larger surface area of catalyst in the form of coating, which ultimately traps more dye molecules, and the rate of degradation increases. Additionally, the inclusion of $SiO_2$ in the coating limits the particle size of $TiO_2$ in the cement and increases the surface area of the catalyst, hence contributing to overall activity. The facilitation of the sol–gel technique can provide control synthesis of $TiO_2$ nanoparticles; however, the generation of low crystallinity products or amorphous products is a challenge. It can be resolved by subsequent heat treatment for crystallization.

Elena et al. reported the advantages of annealed $TiO_2$ as filler in the cement matrix [44]. Mechanical properties of long-aged samples were improved with the addition of annealed $TiO_2$. The prepared composites induce almost 77% degradation of methylene blue (MB) under UV light irradiation. This makes it less effective than a suspension of particles in solution. To overcome this issue, applying the photocatalyst as a coating rather than as filler was advantageous in this case as well. Song et al. proposed a smear method to coat $TiO_2$ on the concrete surface [45]. A detailed study of wettability behavior, as well as photocatalytic activity, was performed. Methylene blue (MB), methyl orange (MO), and rhodamine B (RhB) were considered as model pollutants. The prepared $TiO_2$ paste–cement composite manifested excellent photocatalytic degradation activity as almost 100% of all three dyes were degraded within 50 min. The cyclic performance of the composite was also promising for ten cycles.

Furthermore, the surface of the composite was hydrophobic, with a contact angle of 145°. Hydrophobicity is often seen as the reason why cement and clay materials can be very efficient in removing organic contaminants as they adsorb the molecules before photocatalytic degradation [46]. This is further discussed in the last section. After $TiO_2$, ZnO has been the second-most studied for photocatalytic degradation reactions. Especially, different morphologies of ZnO were studied for the degradation of organic contaminants and for antimicrobial disinfection. It is well known from the literature that different morphologies of photocatalyst can manifest different photocatalytic activities. The photocatalytic efficiency of photocatalytic building materials can be enhanced by growing different morphologies on their surface. Such work was reported by Singh et al., wherein ZnO nanoneedles were fabricated on the surface of cement mortars [47]. The mixing of the ZnO needles in the cement matrix not only provided photocatalytic nature to the cement but also imparted hydrophobicity and antimicrobial characteristics to the cement. The photocatalytic activity of cement was measured by degrading the rhodamine 6G dye under UV light irradiation. The photocatalytic degradation performance of cement ZnO composites in various proportions (0%, 5%, 10%, and 15%,) is shown in Figure 6. As expected, the maximum activity was recorded for the 15% sample under UV light irradiation.

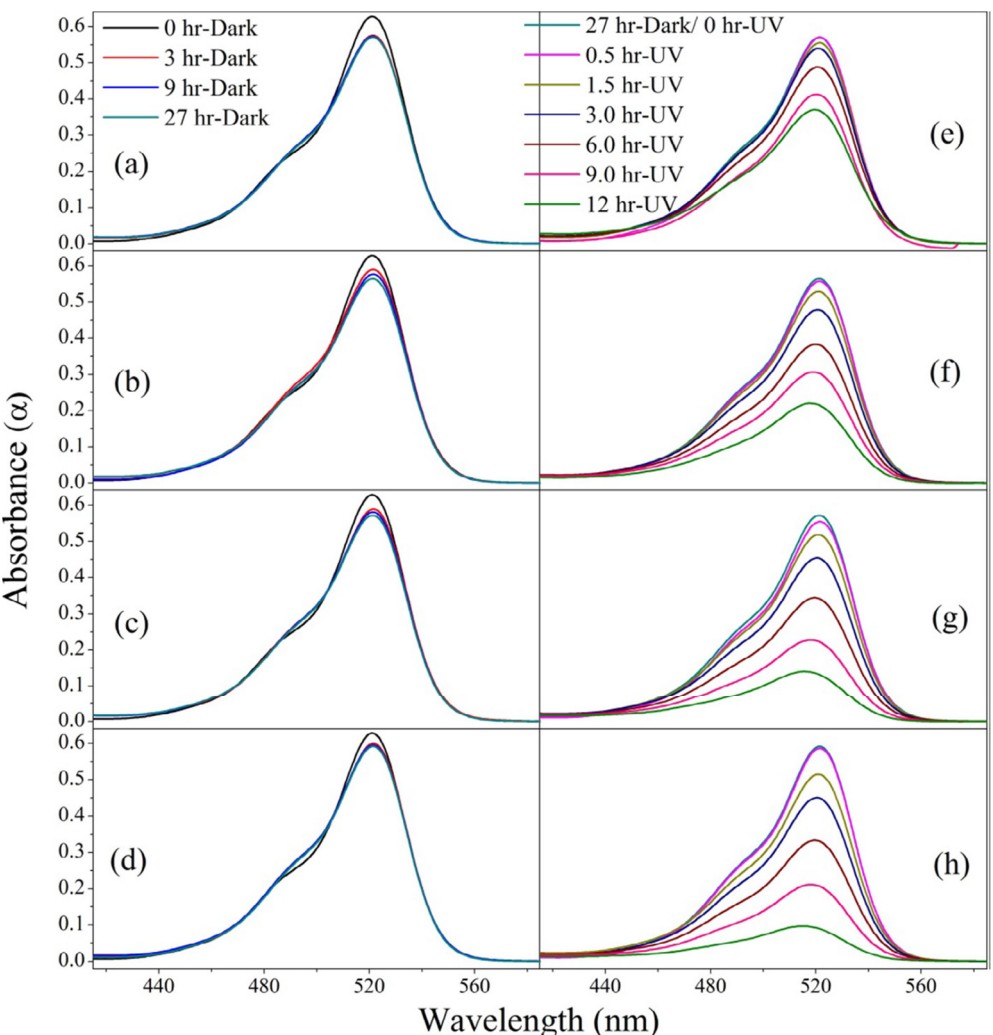

**Figure 6.** (**a**–**d**) UV absorption spectra for the photocatalysis of rhodamine 6G under dark and (**e**–**h**) under UV irradiation on 0%, 5%, 10%, and 15% cement–ZnO(%) composites. Ref. [47].

A more effective approach of growing nanoneedles of ZnO over civil engineering structures (red concrete pavement and gray floor tiling) using the hydrothermal method was proposed by Pivert et al. [48].

The greatest advantage of this method is that the photocatalytic was not used as a filler; rather, it remained on the surface of the structure. Therefore, more surface of the catalyst is exposed to the pollutants as well as to the light and ultimately higher efficiency was observed. Three dyes MB, methyl orange (MO), and acid red 14 (AR 14) were used in the present study. Both MB and AR14 were degraded within two hours; however, slightly higher time was required for MO (3 h). In the previous report, wherein ZnO–nano needles were used as filler, the time taken for MB degradation was 12 h. It indicates that the catalyst in the coating form is significantly more effective than that of filler in the cement matrix. As discussed in a previous section, the efficiency of photocatalyst can be enhanced by delaying the recombination of electron and hole pairs. There are numerous methods to achieve this. However, the use of surface charges of ferroelectric materials is one of the simplest and effective techniques. Additionally, most of the photocatalytic fillers used in the cement–matrix needed UV light to initiate and propagate the reaction. Therefore, only a 3–4% part of the solar irradiation can be used for this purpose. To improve this, the catalyst's bandgap can be shifted slightly into the visible region using doping, structural orientation, formation of hybrid composites, photosensitization of surfaces, etc. [49]. Therefore, there is a huge

scope for the exploration of this photocatalytic filler, which can use a visible light spectrum for catalytic degradation reaction.

One of the prominent issues that cause low efficiencies in photocatalytic degradation reaction is recombination of electron and hole pairs. Feng et al. proposed Z-scheme $TiO_2/g\text{-}C_3N_4$ heterostructure embedded in cement paste to overcome this issue [50]. The prepared composite has a broad light absorption range and low recombination of induced electron hole pairs. The $TiO_2/g\text{-}C_3N_4$ powder has ~97% degradation efficiency of RhB dyes, after 40 min irradiation. Further, the preparation of cement and photocatalytic filler can be done in following ways: (i) direct mixing of cement and photocatalyst in the powder form; (ii) create catalyst slurry and pour it into cement blocks; and (iii) sprinkle catalyst particles on cement slurry. There are some disadvantages of these methods, which include weak binding ability, low photocatalytic efficiency, and less uniform dispersion. To overcome these issues, Wu et al. added Ag in $TiO_2$ and prepared photocatalytic cement mortar by mixing it with cement in various proportions [51]. Samples were fabricated by direct mixing and spraying of the photocatalyst onto the cement. Maximum degradation rate of methyl orange was recorded at 95% for 2 wt.% $Ag\text{-}TiO_2$ sample. Additionally, adding $TiO_2$ in the cement reduced the compressive strength of the cement mortar by 9%; however; no such decrement was recorded with the spraying method. Zhou et al. proposed photocatalytic concrete having $k\text{-}g\text{-}C_3N_4$ as coating on the concrete [52]. The photocatalytic performance of the concrete was tested under water in sunlight irradiation. The degradation rate of methylene blue was observed to reach up to 80% reduction in 30 min. The best performance of the photocatalytic concrete was observed in the 6.5 to 8.5 pH range. In another report, $MoS_2$ nanoflowers were a prepared with cement to from a paste for dye degradation and improve the hydration rate. The report suggested that addition of small proportion of $MoS_2$ nanoflowers can impart photocatalytic activity in cement in sunlight [53].

Another promising approach was demonstrated by Vaish et al. in their report, wherein reduced graphene oxide modified $BaTiO_3$ was used as a photocatalytic filler in Portland cement for visible-assisted water detoxification [54]. $BaTiO_3$ is a popular ferroelectric material that can also be used as a photocatalyst. The inclusion of reduced graphene oxide (rGO) serves two purposes: that of photosensitizer and as an electron reservoir, thereby aiding and promoting the catalytic activity of the ferroelectric. Composites were made in the form of pellets having 5, 10, 15, and 20 percent photocatalytic filler. Rhodamine b was used as a model pollutant, and more than 80% removal was recorded with the 20% filler sample under visible light irradiation. Additionally, the mechanical properties were analyzed with the help of the Rockwell hardness number. The hardness of the samples was found to decrease as the content of filler increased. However, the decrease was marginal. From this discussion, it can be clearly stated that the inclusion of photocatalytic in the form of either filler or coating can impart or enhance the water detoxification capabilities of building materials such as cement, mortar, clay, etc. Recently, piezocatalysis has also emerged as an effective technique to degrade organic pollutants from water. Similar to photocatalysis, the input energy to create the necessary catalysis effect is mechanical vibration. $BaTiO_3$ is one of the most studied piezocatalysts. Sharma et al. mixed $BaTiO_3$ with cement in various proportions and studied the degradation of various organic pollutants. Almost 100% degradation of all pollutants was claimed with 30% $BaTiO_3$ sample [55]. Various functional building materials and their performance in water cleaning are listed in Table 1.

**Table 1.** Performance of various cement-based photocatalytic composites.

| Cement-Based Sample | External Source | Model Pollutant | Catalysis Performance | Ref. |
|---|---|---|---|---|
| Semiconductor ($TiO_2$ and Hombikat UV 100) modified cement | UV light | Atrazine (2-chloro-4-(ethylamino)-6-(isopropyl-amino)-s-triazine) | 7.8% degradation at 7 h | [7] |
| $TiO_2$/cement composites prepared by a smear method | UV light | Rhodamine B (RhB), methylene blue, and methyl orange | Almost 100% in 50 min | [33] |
| $TiO_2$ with a cement binder | UV light | Yellow 17 dye | 60–95% degradation at 8 h | [38] |
| Ti-Zn-Al nanocomposites with cement-based mortars | UV light | Methylene blue (MB dye) | 28% degradation at 3.5 h | [39] |
| $TiO_2$–cement paste | UV light | Methylene blue (MB dye) | - | [40] |
| $TiO_2$/autoclaved cellular concrete composite | UV light | Indigo carmine dye | 100% in 350 min | [41] |
| $TiO_2$-$SiO_2$ nanohybrid based cement materials | UV light | Malachite green oxalate (MG), methylene blue (MB), and methyl orange (MO) | 87% degradation at 120 min | [43] |
| Sol–gel $TiO_2$ nanoparticles for photocatalytic cement composites | UV light | Methylene blue (MB) | 76.60% | [44] |
| ZnO nanoneedle-based cement composite | UV light | Rhodamine 6G | k = 0.147 min$^{-1}$ at 12 h | [47] |
| $BaTiO_3$-rGo composition with Portland cement | Visible light | Rhodamine B | k = 0.4 min$^{-1}$ at 3.5 h | [54] |
| $TiO_2$-containing cement past and mortars | UV light | Rhodamine B | 80% degradation at 7 h | [56] |
| $TiO_2$/$SiO_2$ surface layers on cement panels–plates | Daylight | Red wine | - | [57] |
| $TiO_2$ (P25) deposition on white cement | UV light | Nitrobenzenesulfonic acids | ~0.02–0.03($10^{-5}$ M h$^{-1}$) | [58] |
| $TiO_2$ (P25)-based cement composite | UV light | Benzene | Specific rate of $CO_2$ (650 $CO_2$/ppm) at 80 min | [59] |
| $TiO_2$ fixed on concrete | UV light | 4-chlorophenol | Rate constant 0.277 (mg/(L h)) | [60] |
| $TiO_2$ on pumice stone | UV light | 3-nitrobenzenesulfonic acid (3-NBSA), acid orange-7 | - | [61] |
| $TiO_2$ and ZnO powder mixtures in cement paste | UV light | Methylene blue (MB) | Variable activity with different proportions of $TiO_2$ and ZnO | [62] |
| Cement plates loaded with N,C-modified $TiO_2$ | UV light | RR198 | 49.3% degradation at 100 h | [63] |

### 2.2. Antifungal and Antimicrobial Cement-Based Water Detoxification

Often the storage tanks and canals are made up of cement, concrete, and other building materials. Therefore, it is essential to maintain aesthetic quality for these structures. The extended exposure of the water on these surfaces can often cause algae and fungal growth on these structures. There are two reports which suggest that the inclusion of photocatalytic filler can induce antimicrobial characteristics to the cement. Singh et al. studied the antimicrobial performance of ZnO–cement composites. Antimicrobial studies were performed using bacterial strains Escherichia coli (*E. coli*) (JM109, Promega Gram-negative), Bacillus

subtilis (MTCC121, Gram-positive), and fungal strain Aspergillus niger (MTCC281) for all the composites [47]. Figure 7a shows the impact of various compositions of prepared ZnO–cement composites on the growth of the bacterial colony of *E. coli* and *B. subtilis* under dark and solar irradiation.

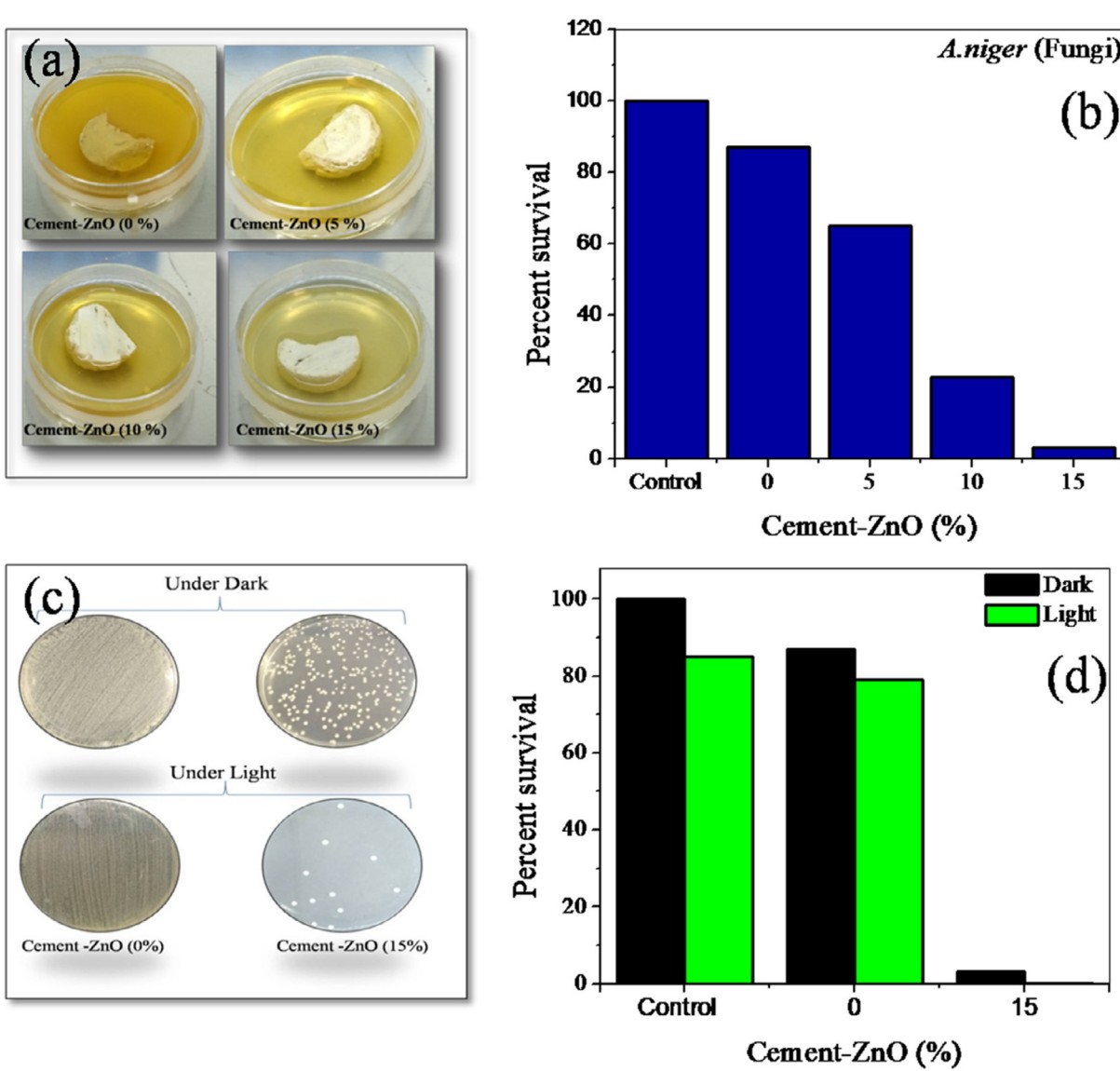

**Figure 7.** (**a**) Cement–ZnO composite bacterial disinfection *E. coli* and (**b**) *B. subtilis*. (**c**) Pictures of colony-forming *E. coli* on agar plate under dark and sunlight. (**d**) Percentage survival of bacteria on cement–ZnO composite was improved under sunlight (ref. [47]).

It was observed that bacterial growth fell as the percentage of ZnO increased in the cement. Furthermore, the experiment of colony-forming of *E. coli* shows suppressed activity with 15% ZnO-cement. Solar-light-assisted cement–ZnO composite displayed almost no growth of fungi at 15% cement–ZnO composite compared to that of control and pure cement under 2 h sunlight irradiation. The mechanism proposed for this antimicrobial activity was the generation of reactive oxygen species, which induce oxidative stress in bacterial cells. Similarly, Nasim et al. also demonstrated the antimicrobial performance of nanoTiO$_2$ and cement composites. Photocatalytic inactivation of *E. coli* by cement samples containing 1, 5, and 10 wt.% TiO$_2$ was investigated under UV irradiation. Significant bacteria removal was observed with 5% and 10% TiO$_2$ samples.

It is clear from the above discussion that the two most prominently used photocatalysts, TiO$_2$ and ZnO, can induce antimicrobial characteristics in cement. A work also investigating

the mechanical properties found an improvement in strength in cement–photocatalytic filler composite with the addition of silica nanoparticles. Nanosilica can facilitate nucleation by acting as a pore filler and promotes the pozzolanic reaction. Sikora et al. reported mechanical and bacterial disinfection properties of silica/titania core–shell nanocomposite in the cement. A significant improvement in compressive strength (15%) was recorded with the $SiO_2/TiO_2$–cement sample [32]. In another report, Sikora et al. presented a more detailed study of antibacterial cement composites after the inclusion of $Al_2O_3$, CuO, $Fe_3O_4$, and ZnO nanoparticles in the cement [64]. All of the nano-oxides showed significant antimicrobial activity up to a certain level. However, the microorganisms were able to regrow after a certain time interval. Four different bacterial solutions: *E. coli*, *S. aureus*, *P. aeruginosa*, and *C. albicans*, were tested. ZnO was found most effective, followed $Fe_3O_4$ and CuO in the *E. coli*, *P. aeruginosa* suspensions. $Fe_3O_4$ was most effective in the *C. albicans* microbial suspension. Note that the toxicity of these nano-oxide–cement composites was not permanent as most of the cultures were able to regrow after 24 h of incubation. The adherence test of biofilm formation showed that the inclusion of nanoparticles in the cement diminish the formation of biofilm in the studied bacteria. Further, eggshell-waste-derived CaO–cement composites were also found to have antibacterial properties. The eggshell was heated to 900 °C to obtain CaO, which was further mixed with mortar in different proportions. The best results were obtained with the 15% CaO sample [65].

Mixing of $TiO_2$ in cement can also be used for algae inhibition. It was reported that photocatalytic films of $TiO_2$ and $WO_3$ over cement substrate were able to reduce algae growth by up to 66% more than that of uncoated samples [66]. Further improvement to 88% could be obtained by adding 1 wt.% of noble metals such as Pt or Ir. This discussion concludes that photocatalytic cement-based materials can be effective not only for the removal of organic contaminants but also for bacterial disinfection in water. However, there is enough scope of improvement before the commercial use of these composites for water cleaning. This is elucidated in the last section.

### 3. Adsorption-Based Water Detoxification

Similar to photocatalysis, adsorption is also one of the commercially used techniques for water detoxification. This technique is based on the accumulation of the pollutants on the surface of adsorbent extracted from the solution. Until now, many adsorbents have been tested and applied to water purification, such as activated carbon, clay, zeolites, etc. However, activated carbon has been the adsorbent of choice in commercial applications due to its higher surface area and large adsorption capacities [67–69].

The adsorption process offers some advantages over the photocatalytic process. For example, unlike photocatalysis, it is independent of the lighting conditions of the system. Another major advantage is the ability to remove toxic heavy metals from water. On the other hand, the biggest drawback is the nonreusability of the adsorbent. A typical adsorption process starts rapidly, and as the adsorption sites keep filling, the rate of adsorption slows. It completely stops after all adsorption sites are saturated. There are a few strategies to recycle respective adsorbents, but these are often complex or not efficient. However, as mentioned in the previous section, hydrophobicity can lead to the adsorption of organic molecules. Coupling this with a photocatalytic degradation process can make this an efficient approach to remove organic contaminants. Although the building materials and their composites have not been much explored for water remediation, there are a few reports available in the literature which demonstrate water cleaning using these materials. For example, cement kiln, a solid waste generated from cement industries, was applied as an adsorbent for dye removal [70].

The high surface area and porous structures of zeolites and carbonaceous materials such as activated carbon can be combined with cement, and the composite can be used for detoxification of water. It is quite significant from the available literature that most of the cement composite materials for water remediation are fabricated with photocatalytic materials. Heavy metal (Hg, As, Pb, etc.) contamination cannot be treated with a catalytic

process. This can be resolved by applying adsorption-based water detoxification using composites of adsorbents and building materials such as cement. Yang et al. reported the removal of heavy metal ions from the water with a mixture of zeolite and Portland cement [71]. Zeolite is a naturally occurring adsorbent with a large surface area and excellent metal ion removal capacity [72]. In this investigation, the zeolite and cement were mixed in a proportion of 3:1 using a vacuum extruder. The prepared samples were aged for 30 days in water at ambient conditions. The prepared composite (ZeoAds) was found to successfully adsorb Cu, Cd, Pb, and Zn metal ions. The effect of solution pH on the adsorption was also significant. The higher pH favors the adsorption due to precipitation of metal complexes. The maximum adsorption capacities of Cd, Zn, Cu, and Pb were recorded 10.87, 12.85, 23.25, and 27.03 mg/g, respectively. The adsorption capacities of ZeoAds for various metal ions were also compared with that of activated carbon. They found significantly higher efficiencies for ZeoAds than for activated carbon, illustrating the high potential of this approach.

Among all heavy metal contaminants in water, arsenic (As) is one of most toxic and undesirable. In many countries such as Bangladesh, India, Mongolia, Taiwan, and Vietnam, the level of arsenic exceeds the permissible limits of the World Health Organization (WHO). The intake of water contaminated with arsenic can cause serious diseases such as lung, liver, and bladders cancer. Therefore, it is vital to remove such harmful metal ions from water. Gupta et al. reported the removal of as from water using iron oxide-coated cement (IOCC) in a fixed bad reactor setup [73]. The iron oxide was coated using the dip-coating method on the cement mortars. The cement samples were dipped in the solution of ferrous nitrate solution for sufficient time, followed by furnace drying at 100 °C for 16 h. The characteristic parameters of adsorption (adsorption rate, adsorption capacity, depth of exchange zone) were obtained using the Bohart and Adams sorption model. The maximum adsorption capacity of arsenic removal was 505.3 mg/g of adsorbent. Additionally, the flow rate through the column affects the removal efficiency. More removal was observed at higher flow rates. For an effective sorption process, the reusability of adsorbents is essential. As mentioned, a desorption process is necessary for recycling and reactivation. In this report, 10% NaOH solution was used as desorption media. The removal of arsenic was recorded at 99% with 13-bed volumes of NaOH. After desorption, the adsorbent retains almost 90% performance. NaOH treatment is, of course, rather drastic and complex for industrial-scale applications. For the adsorption of organics, other methods can be facilitated as well. In this case, thermal degradation, washing with organic solvents, and biological treatment were shown to be successful for clay-derived material but still quite challenging [74].

A highlighting point of adsorption technique in water remediation is the variety of adsorbents and their sources. Especially, the conversion of various wastes such as fly ash, rice husk, peanut hull, and other agricultural and industrial waste not only explore low-cost adsorbents but also contribute effective waste management [75]. There are few reports in the literature in which adsorbents derived from waste were mixed with cement or concrete, and effective water cleaning was reported. Pimraksa et al. derived zeolite and zeolite-like material from bottom ash (BA) obtained from a lignite power plant [76]. It was mixed with cement mortars and heavy metal encapsulation. Two different types of zeolites were obtained with the combination of bottom ash and KOH solutions. Natrolite l-k zeolite (NAT-K) was prepared with bottom ash and 7 M KOH, a zeolite-like material (KASH), were obtained with fine bottom ash powder and 9 M KOH solution. These adsorbents, NAT-K, KASH, and bottom ash, were mixed in a proportion of 05, 10, 20, and 30% by weight in the Portland cement to produce composite materials for heavy metal encapsulations. Both NAT-K- and KASH-filled cement mortars were found to remove more than 95 percent of heavy metal ions (chromium, nickel, and cadmium) from their aqueous solution. The mechanical properties were also influenced by the inclusion of adsorbents. The strength and density of cement mortars marginally improved by 5% NAT-K cement mortars due to the pozzolanic and microfiller effect. For 10% NAT-K samples, the compressive strength of the composite was almost identical to the pure cement samples. Similarly, KASH can also be added in the

cement as filler up to 10% without any significant reduction in the compressive strength of the cement mortars. Carbon composites can also be used for the removal of organic pollutants from the water. In a recent report, diesel exhaust emission soot was mixed in the Portland cement, and effective dye and detergent cleaning were demonstrated [77]. Diesel exhaust emission soot (DEES) is a waste generated from automobiles by burning diesel and other gasoline fuels. Being a carbon-rich material, DEES possesses good adsorption capacity. DEES was added to the cement in different proportions (maximum 50%), and the coatings of DEES-cement were fabricated. The prepared coatings are found to successfully adsorb a variety of pollutants from the water, such as MB, rhodamine b, ciprofloxacin, and detergent. The adsorptive removal of MB is reported in Figure 8.

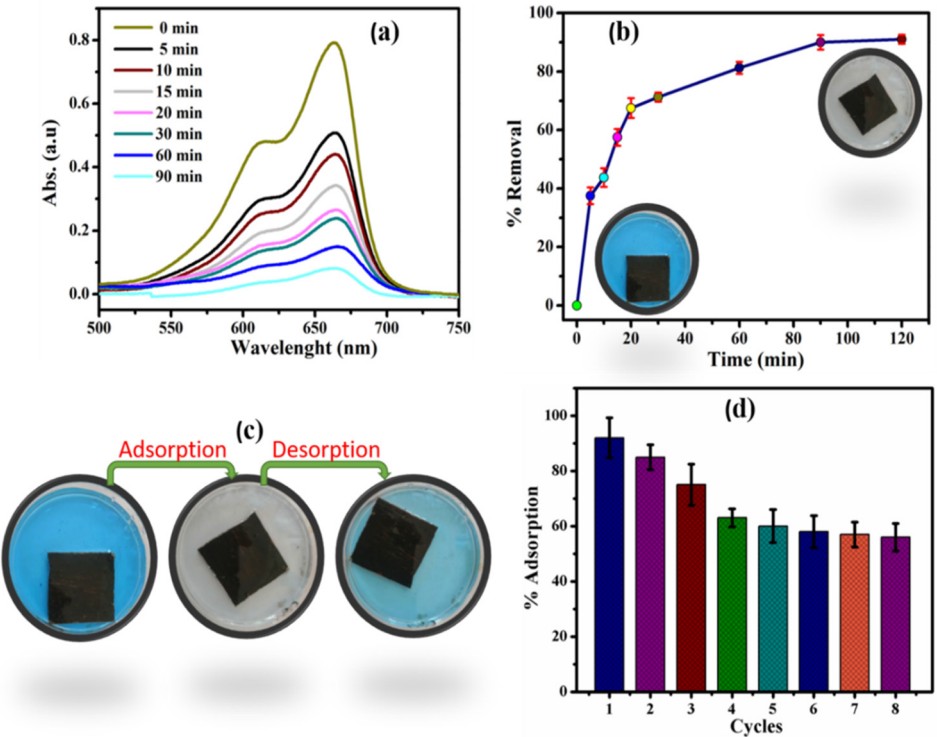

**Figure 8.** (**a**) Absorption vs. wavelength spectra for MB. (**b**) Adsorption% of MB through time. (**c**) Adsorption–desorption of MB and (**d**) cyclic performance of cement–DEES coatings (ref. [77]).

The coatings can be reused for six cycles after dissolving the adsorbent in the desorption media of ethanol. Additionally, the detailed study with the help of some standard kinetic models suggests that the kinetics of the adsoption process can be best explained with the help of a pseudo-second-order kinetic model. Additionally, the mechanism of diffusion of the dye into the pores of adsorbent was explained with the help of an intraparticle diffusion model. According to this model, if the curve passes through the origin, intraparticle diffusion will be the only rate determining step. However, in the study, the intraparticle diffusion curve did not pass through origin, which suggests that the intraparticle diffusion is not the only rate determining step during adsorption of MB on the cement-DEES coatings. The adsorbed dye was desorbed in the desorption media and it shows that adsorption was physiosorption [77]. The study clearly shows that the inclusion of carbonaceous material in the cement can impart adsorptive properties to the cement, and it can be used for water detoxification in single-component systems. However, in the real world, pollutants are found mostly in a multicomponent system. In one of the reports, the cement–carbon composite was prepared and used as an adsorbent in both single and multicomponent systems [78]. Three dyes, brilliant green (BG), methyl orange (MO), and methyl blue (MB), were taken as adsorbates. For binary systems combination of two and in the ternary system, all three were mixed. The maximum adsorption capacity of

cement–carbon composite (CCC) was determined by fitting suitable isotherm models. The maximum adsorption capacities of CCC toward the single, binary, and multicomponent system is shown in Figure 9.

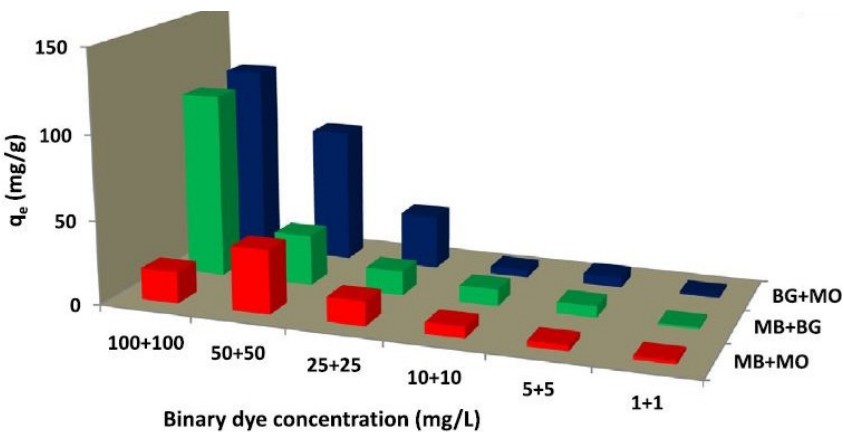

**Figure 9.** Removal in binary dye adsorption system using CCC (ref. [78]).

The results indicate that CCC can effectively adsorb dye in both single and multi-component systems. The adsorption capacity was found in the order BG > MO > MB for CCC and BG + MO > MB + BG > MB + MO for binary adsorption. After adsorption, the desorption study was also performed, and up to 40% desorption was achieved. The above discussion and data indicate that adsorbent can be used as filler in the cement, and adsorption-based water detoxification can be achieved. The other highlighting point here is the use of waste-derived adsorbents in combination with cement and other building materials, which also offer effective waste management. However, cement kiln dust (CKD) can be used directly as an adsorbent. The first report in this context was presented by Sami et al., wherein adsorption of heavy metal ions (Cr, Fe, Co, Cu, etc.) was studied using cement kiln dust [79]. Effect of various process parameters such as time of contact, dosage of adsorbent, and pH was also studied in detail. It was found that cement kiln effectively removes a variety of metal ions from the water with almost 100% removal of chromium ions. It is clear from this preliminary report that CKD can be used as an adsorbent. A more detailed study exploring the potential of CKD was presented by Mahmoued et al., wherein real-time samples from the effluent of the textile industry in Egypt were taken for the experiments [80]. The study records laboratory-scale experiments to test the efficiency of cement kiln dust (CKD) and CKD + Coal filters in removing color, turbidity, organic substances, and heavy metals from textile wastewater. It was found that samples collected from the effluents of the textile industry have lower chemical oxygen demand (COD) after treatment with CKD and coal filters. The experiments were performed at different hydraulic-loading levels. The maximum removal of 97% of color, 76% of turbidity, 84% of COD, and 77% of BOD was obtained with CKD+ Coal filters.

The result suggests that CKD can be used as an adsorbent either alone or in combination with coal for water detoxification for real-time applications. Kinetic analysis of dye adsorption was carried by Magdy et al. [81]. Numerous kinetic models were fitted to the kinetic data of adsorption, and the findings suggest that Elovich and pseudo-second-order kinetic models are the best fit to kinetic data during the adsorption of basic blue and basic red dyes. The adsorption process led to saturation within 90–300 min depending on the adsorbent dose dye and initial dye concentration. Both film and intraparticle diffusion were found to be the rate-affecting step during adsorption. Furthermore, it is well known that process parameters such as time of contact, the dosage of adsorbent, pH, etc., have a significant effect on the effectiveness of adsorbent. Therefore, the optimization of these parameters has great importance in achieving maximum adsorption capacity. Response surface methodology is an analytical technique that can be used as a tool for process pa-

rameter optimization. Elevli et al. used this methodology for process optimization during adsorption of malachite green (a basic dye) [82]. The dye concentration, adsorbent dose, and time of contact were optimized with the help of central composite design (CCD), and optimized values were 110 mg/L, 22 g, and 123 min, respectively. CKD can also be used in combination with other adsorbents to provide better adsorption and mechanical properties. Salem et al. reported a combination of cement kiln dust, zeolite, and bentonite in the form of a Raschig ring, and removal of lead ions was studied [83]. The experimental setup and adsorption capacity of CKD toward lead ions are shown in Figure 10a,b respectively.

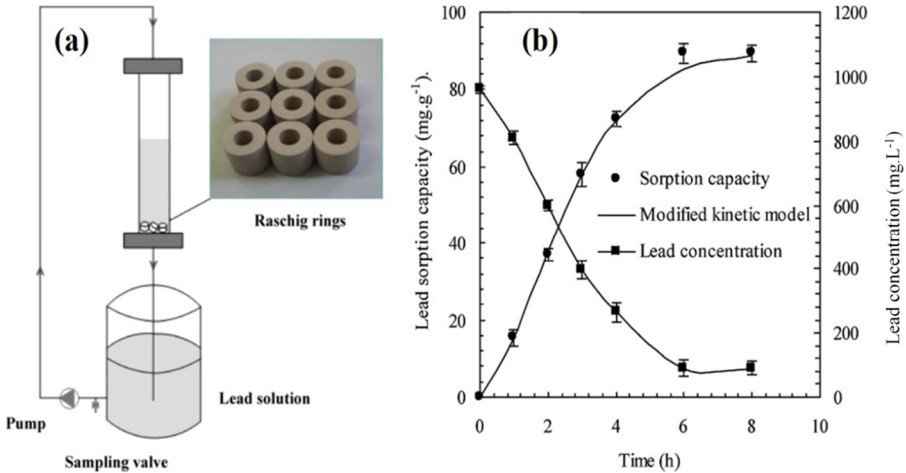

**Figure 10.** (**a**) Experimental setup for lead adsorption on CKD and (**b**) the sorption capacity time data for rings prepared with optimum composition (ref. [83]).

A comparative study of various cement-based composites used as an adsorbent is shown in Table 2.

**Table 2.** Various cement-based composites used as an adsorbent.

| Cement-Based Sample | Model Pollutant | Adsorbent Performance | Ref. |
|---|---|---|---|
| Zeolite–Portland cement | Heavy metal | 90% of the Cu within 30 min | [71] |
| Iron oxide-coated cement | Arsenic | 786–963.75 | [73] |
| Cement mortars hybridized with zeolite | Heavy metal | 97%. | [76] |
| Cement–carbon composite | Acidic and basic dyes | 21.50, 9.06, and 20.20 mg/g for BG, MB and MO in the single-dye system, respectively | [78] |
| Cement kiln dust | Heavy Metals | 40–99.4% | [79] |
| Cement kiln dust and coal filters | Textile industrial effluents | 97% of color, 76% of turbidity, 84% of COD, 77% of BOD and 94% of $PO_4-3$ from raw textile wastewater | [80] |
| Cement kiln dust | Dyes | BB69 and AR114 were 2119 mg/g and 2125 mg/g, respectively. | [81] |
| Cement kiln dust | Dye | 99.4623% | [82] |
| Mixture of cement kiln dust, zeolite, and bentonite | Lead | 15.5 to 57.8 mg g$^{-1}$ | [83] |
| Iron oxide-coated cement | As(V) | 505.3 mg/L | [84] |
| Cement kiln dust | Acidic wastewater | 87% | [85] |

Other than the presence of toxic contaminants, higher acidity is also one of the issues to be addressed. Acidic waste streams such as acid rock drainage (ARD) from base metal mines contain hazardous metals that, along with high acidity, damage ecosystems. CKD can be used as an acid neutralizer. The above discussion clearly demonstrates that building materials can be admixed with a suitable photocatalyst, adsorbent, and other nanomaterials to impart environmental remediation characteristics to them. Along with the presence of various organic and inorganic pollutants, oil spills are also one of the serious issues of water contamination [86].

Unique wettability materials have always been the first choice to counteract this problem. Cement-based composite can be made hydrophobic/hydrophilic by incorporating suitable filler or coatings. However, most of these composites were made for self-cleaning purposes. Recently, Song et al. demonstrated cement-coated steel mesh for oil–water separation [87]. The superoleophobic mesh was fabricated using the dip-coating method. The copper mesh was dipped in the cement paste for 30 s and pulled out at a speed of 0.01 m/s. Numerous oils (hexane, diesel, peanut oil, lubricating oil, and dichloromethane) with a wide range of kinematic viscosities were mixed in the water and tested for oil–water separation efficiency of the mesh. Promising separation efficiency of 94% and above was obtained all oil-water emulsions.

Additionally, the prepared copper–cement mesh was found reusable for 30 cycles. Therefore, from the above discussion, it is clear that cement-based catalysts and adsorbents can be made environmentally friendly by incorporating suitable photocatalyst or adsorbent as filler or coating.

## 4. Discussion of Limitations and Future Scope

The present report gives an overview of the functionalities of cement and clay-based composites concerning water treatment for contaminant removal. The discussed research is often clearly focused on wastewater treatment, but sometimes the capability to implement the composites in buildings is also highlighted. This should then lead to the opportunity to develop self-cleaning surfaces or the contribution of constructions such as pillars of bridges toward water cleansing. However, it is very often difficult to judge whether such applications are actually feasible. In the authors' opinion, it would be beneficial to address the following points in the future to clarify the direction in which the research is heading and to elucidate its full potential.

a.  Photocatalysis is the most exploited technique for water treatment using functional building materials with $TiO_2$ and ZnO being primary photocatalyst. Both catalysts mostly require UV light irradiation for their effective usage. However, solar irradiation only contains 4% UV irradiation, which significantly restricts photocatalytic-based functional building material applications for storage tanks and canals, etc. One way to circumvent this issue is the use of doping, structural orientation, formation of hybrid composites, and photosensitization of surfaces for bandgap tuning. The visible-light-active catalyst can be prepared using these methods, and visible-light-active photocatalytic building materials can be fabricated for practical applications. However, that does not address the aspect that the efficiency of the photocatalyst depends on the intensity of the light reaching the surface of the catalyst. In most reports, the catalyst was used as filler in the cement matrix, which ultimately lowers the proportion of light falling on the catalysts surface, and the efficiency of degradation is diminished. There are different solutions proposed by various researchers to encounter this problem. The porous $TiO_2$/foam composites were fabricated by mixing foaming agent. The porous structure increases the surface area of the catalyst, and more light penetration in the cement can be achieved. Similarly, floating cement–$TiO_2$ composites were prepared to improve light irradiation on the surface of the catalyst. Another approach studied by few researchers was to use catalyst as coatings. It maximizes the light irradiation proportion on the surface of the catalyst, and higher degradation efficiency was observed. These are strategies beneficial for

an application in direct wastewater treatment. However, water with high turbidity may generally prevent the application of this approach in structures such as pillars from bridges over lakes and rivers. Thus, the self-cleaning properties or contribution to water cleaning of the structures may be inhibited.

b.  In case the claim is made that investigated composites are useful for self-cleaning surfaces in constructions, data should generally be provided that the mechanical properties do not suffer or could even be enhanced. The filler in the cement can alter the hardening and the mechanical properties of cement. There are few reports which have studied the influence of these fillers on aging and mechanical properties. Still, more robust and detailed studies are required to analyze the most optimum proportion of filler to obtain the best catalytic activity without much affecting its mechanical and hydration properties. Additionally, the robustness of coatings is essential to ensure longevity in extreme weather conditions.

c.  Very often, the merit of using cement-based composites is just the effective removal of contaminants and not necessarily the applications in buildings and constructions. This should be highlighted rather with the focus on efficient and economic wastewater treatment plants.

d.  Leaching of the catalytically active nanomaterials into the environment is critical, as well. Therefore, the environmental impact of the composites themselves should also be in the focus of the research. In some publications, the conclusion is drawn that coatings on cement are better than embedding the catalysts in the cement. Even though this may be true for many approaches, leaching of the nanomaterials is likely much more prominent for coatings. Furthermore, detrimental effects could occur even in the case of sufficiently bound nanomaterials. The removal and degradation of molecules and bacteria are not specific. Thus, the use in constructions and buildings could, for example, lead to an undefined impact on the microbiome of the surroundings. Additionally, harmful degradation products could also develop, which are as toxic as or even more toxic than the contaminant itself.

e.  Economic aspects of fabrication and the implementation of these functional materials have not been discussed in any of the reports. It is just assumed that the strategies are more economical because they are based on cement. However, no definite proof or comparison with other procedures is given. Therefore, even if these functional materials found suitable for performance background, it is vital to perform an economic study for the feasibility of practical applications.

f.  Adsorption-based functional building materials are more versatile than photocatalytic building materials as they can adsorb both organic and inorganic pollutants (heavy metals ions). However, the biggest disadvantage of adsorbents is nonreusability. Unlike photocatalysis, pollutants are not degraded in the adsorption process. They are transferred to the surface of the adsorbent. Therefore, the reusability of adsorbent is possible after desorption of adsorbed compounds in a suitable desorption media. It may make the process complex and uneconomical for practical applications.

g.  Ferroelectric materials can be explored for better catalytic efficiency due to their spontaneous polarization. Recent reports suggest that remnant polarization on the surface of ferroelectric materials can generate ROS in the water, and bacterial disinfection can be achieved. Composite of cement and ferroelectric materials can be studied for photocatalytic and piezocatalytically degrading organic contaminants and bacterial disinfection.

**Author Contributions:** Conceptualization, R.V. and V.P.S.; methodology, E.S.Y.; writing—original draft preparation, V.P.S.; writing—review and editing, V.P.S.; supervision, R.V.; funding acquisition, E.S.Y. All authors have read and agreed to the published version of the manuscript.

**Funding:** This research received no external funding.

**Data Availability Statement:** The raw/processed data required to reproduce these findings cannot be shared at this time due to technical or time limitations.

**Acknowledgments:** The authors express their appreciation to "The Research Center for Advanced Materials Science (RCAMS)" at King Khalid University, Saudi Arabia, for funding this work under grant number RCAMS/KKU/002-22.

**Conflicts of Interest:** The authors declare no conflict of interest.

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
