# Peer review of "A Review on Cement-Based Composites for Removal of Organic/Heavy Metal Contaminants from Water"

_catalysts, doi:10.3390/catal12111398_

Round 1

Reviewer 1 Report

Congratulations on your manuscript, and especially for the effort put into the graphic part of it.

Author Response

Reviewer 1:

Congratulations on your manuscript, and especially for the effort put into the graphic part of it.

Thanks for your kind words.

Reviewer 2 Report

The authors prepared good manuscript, but it needs revision. As this is a review paper, it is better the authors mention about the results of other researchers clearly and shortly, the text of manuscript is too long. Also, the writing level of manuscript is poor. I suggest major revision.

Some of my comments for authors:

1.      The text of manuscript is too long. Please delete some unnecessary parts of manuscript.

2.      The quality of most figures is very low. I suggest the authors to improve the quality of figures significantly. This manuscript is a review article, then it is better the authors try to merge some figures of different references in one figure instead of adding so many figures. Also, the authors must add reference for all the figures if they did not draw with themselves.

3.      Fig. 1 is very simple scheme. Please delete it or insert a good scheme to show the sources of water pollutants or …, It does not show important data.

4.      In Figure 4. Please remove the scheme related to production of ROS, which you used it before in Fig.1. ROS is common and everyone know about its meaning. Also it is better you show the abbreviations (labels inside the square for cellular DNA, ROS and … ) beside the scheme not in center.

5.      Page 7: These sentences need revision.

For water purification purposes, it is thus important that along with parameters like the selection of catalytic materials, amount of catalyst used, etc.,

one can also adjust other reaction parameters like pH of the solution to achieve better efficiency with the same material.

6.      The authors did not mention about the Fig.9 d in figure caption.

7.      Figure 11 and Figure 12 are not clear, I suggest remove these figures.

8.      Please revise Table 1 and Table 2 completely. Please unify the reported results and insert references with full information. In catalyst performance, for some sample you wrote their efficiency, for other samples the rate of degradation and …. It is not clear. Please add some columns for degradation efficiency, reaction rate and time of process, then complete with data of references, Same to Table 2 , for adsorption efficiency and adsorption time. According the Tables, we want to compare the results for degradation or removal of pollutants, when the time of process is not mentioned, then we cannot compare the results exactly. Please check recently published review articles in this field and check the Tables.

9.      The cited references in text of manuscript before Table 1 are till 55, but in Table 1 there are some references like 79, 80, ….86,87, and then in text of part 2.2, the references are from 47… I suggest please check the references order completely. The citation in text and Tables must be ordered. Also, same problem is in Table 2 reference 87,89, … However, the final reference in text of manuscript and Fig. 17 is 71.

10.  Most of the cited references are very old. In cited references, I found less than 6 articles published in 2021-2022. Please insert some new articles (2021-2022) and update the information of the manuscript.

Author Response

Reviewer:

The authors prepared good manuscript, but it needs revision. As this is a review paper, it is better the authors mention about the results of other researchers clearly and shortly, the text of manuscript is too long. Also, the writing level of manuscript is poor. I suggest major revision.

       Some of my comments for authors:

  1. The text of manuscript is too long. Please delete some unnecessary parts of manuscript.

Some part of the manuscript has been deleted.

  1. The quality of most figures is very low. I suggest the authors to improve the quality of figures significantly. This manuscript is a review article, then it is better the authors try to merge some figures of different references in one figure instead of adding so many figures. Also, the authors must add reference for all the figures if they did not draw with themselves.

Thanks for highlighting the point. We have now removed some figures of low quality.

  1. 1 is very simple scheme. Please delete it or insert a good scheme to show the sources of water pollutants or …, It does not show important data.

Figure 1 has been removed.

  1. In Figure 4. Please remove the scheme related to production of ROS, which you used it before in Fig.1. ROS is common and everyone know about its meaning. Also it is better you show the abbreviations (labels inside the square for cellular DNA, ROS and … ) beside the scheme not in center.

Figure 4 has been modified as per your suggestion.

  1. Page 7: These sentences need revision.

For water purification purposes, it is thus important that along with parameters like the selection of catalytic materials, amount of catalyst used, etc.,

one can also adjust other reaction parameters like pH of the solution to achieve better efficiency with the same material.

          The sentence has been revised now in the revised manuscript.

  1. The authors did not mention about the Fig.9 d in figure caption.

Figure captions are now been mentioned.

  1. Figure 11 and Figure 12 are not clear, I suggest remove these figures.

Figures 11 and 12 has now been removed.

  1. Please revise Table 1 and Table 2 completely. Please unify the reported results and insert references with full information. In catalyst performance, for some sample you wrote their efficiency, for other samples the rate of degradation and …. It is not clear. Please add some columns for degradation efficiency, reaction rate and time of process, then complete with data of references, Same to Table 2 , for adsorption efficiency and adsorption time. According the Tables, we want to compare the results for degradation or removal of pollutants, when the time of process is not mentioned, then we cannot compare the results exactly. Please check recently published review articles in this field and check the Tables.

We agree with reviewer’s point however it has been done because in some of the articles degradation efficiency is mentioned and in some other rate of degradation. Therefore, this configuration has been adopted.

  1. The cited references in text of manuscript before Table 1 are till 55, but in Table 1 there are some references like 79, 80, ….86,87, and then in text of part 2.2, the references are from 47… I suggest please check the references order completely. The citation in text and Tables must be ordered. Also, same problem is in Table 2 reference 87,89, … However, the final reference in text of manuscript and Fig. 17 is 71.

References have been arranged in proper manner in the revised manuscript.

10. Most of the cited references are very old. In cited references, I found less than 6 articles published in 2021-2022. Please insert some new articles (2021-2022) and update the information of the manuscript.

We have included all relevant papers in the revised manuscript from recent two years.

Reviewer 3 Report

The manuscript by Singh et al. reviews the recent development in removing contaminants from water by utilizing cement-based composite photocatalysts. The manuscript contains many typos/errors, which need to be solved for better readability. In addition, the captions are too short and don't explain the images. The figures haven't been explained/discussed well in the text. For a review paper, the reference list is too short as well; I believe that the authors can add more discussions, which will certainly enhance its readability. For example, after this sentence, "With respect to photocatalysis, the most discussed materials are TiO2 and ZnO majorly as nanosized particles[8, 9].", the following refs can be implemented. 

https://doi.org/10.1016/j.cattod.2017.12.008

10.1021/acscatal.9b03322

10.1021/acscatal.0c00556

It is well-known that all the photocatalytic processes and efficiency are mostly determined by fundamental events such as adsorption, desorption, photolysis, etc., which are well-described in the following chapter, which I believe the authors of this paper can also benefit from. https://doi.org/10.1016/B978-0-323-88449-5.00006-1 

Another drawback is a lack of comparison between different systems, which is more important for review papers. 

I suggest that the authors consider the aforementioned comments before my final decision.

Author Response

Reviewer 3:

The manuscript by Singh et al. reviews the recent development in removing contaminants from water by utilizing cement-based composite photocatalysts. The manuscript contains many typos/errors, which need to be solved for better readability. In addition, the captions are too short and don't explain the images. The figures haven't been explained/discussed well in the text. For a review paper, the reference list is too short as well; I believe that the authors can add more discussions, which will certainly enhance its readability. For example, after this sentence, "With respect to photocatalysis, the most discussed materials are TiO2 and ZnO majorly as nanosized particles[8, 9].", the following refs can be implemented. 

https://doi.org/10.1016/j.cattod.2017.12.008

10.1021/acscatal.9b03322

10.1021/acscatal.0c00556

It is well-known that all the photocatalytic processes and efficiency are mostly determined by fundamental events such as adsorption, desorption, photolysis, etc., which are well-described in the following chapter, which I believe the authors of this paper can also benefit from. https://doi.org/10.1016/B978-0-323-88449-5.00006-1 .Another drawback is a lack of comparison between different systems, which is more important for review papers. 

Thanks for all your suggestions and comments. We have now modified the manuscript as per your suggestions.

Round 2

Reviewer 2 Report

Dear Authors,

I suggest acceptance for this manuscript.

Regards

Reviewer 3 Report

The manuscript can be accepted in the present form.